# Insights into COVID-19 and Its Potential Implications for Kidney Dysfunction

Adel Abdel-Moneim [1], Eman H. Bakry [2] and Mohamed Y. Zaky [1,2,*]

1  Molecular Physiology Division, Faculty of Science, Beni-Suef University, Beni-Suef 62511, Egypt; adel_men2020@yahoo.com
2  UPMC Hillman Cancer Center, Division of Hematology and Oncology, Department of Medicine, University of Pittsburgh, Pittsburgh, PA 15213, USA; emanhussein225@gmail.com
*  Correspondence: mohamedzaki448@science.bsu.edu.eg

**Abstract:** Coronavirus disease 2019 (COVID-19), caused by the severe acute respiratory syndrome coronavirus 2 (SARS-CoV-2), has had a significant impact on the world's demographics, resulting in over 6 million deaths globally. COVID-19 has been associated with a variety of disease manifestations in various organ systems, including kidney disease, in addition to pulmonary manifestations. Infection with SARS-CoV-2 can not only cause new kidney damage but also make treatment and care more difficult, as well as increase mortality in people who already have kidney problems. COVID-19 is indeed associated with a variety of renal pathologies, such as acute tubular necrosis, proteinuria, hematuria, and thrombosis complications. Cytokine storms, hypoxemia, direct viral invasion via angiotensin-converting enzyme 2 and cathepsin L, electrolyte imbalance, and fever are among the pathophysiological mechanisms underlying these clinical symptoms. Over the last two years, many COVID-19 vaccines have been discovered. However, there have been a few case reports of AKI, AKD, proteinuria, edema, gross hematuria, and other renal side effects that necessitated hospitalization after receiving COVID-19 vaccinations. Thus, the current review aimed to evaluate COVID-19-induced kidney dysfunction in terms of clinical features, pathogenesis, long-term outcomes, and vaccine harms based on the most up-to-date findings.

**Keywords:** SARS-CoV-2; COVID-19; COVID-19 implications; kidney dysfunction

## 1. Introduction

In December 2019, adults in Wuhan began presenting to local hospitals with serious pneumonia of unknown origin. On January 7, the virus was described as a coronavirus [1]. According to epidemiological data, the severity of the illness from coronavirus disease 2019 (COVID-19) is as high as 25%, and even though the lungs are the main organs affected, the kidney is among the different organs that are significantly affected with the SARS-CoV-2 infection [2,3]. SARS-CoV-2 infection causes systemic symptoms such as heart and liver diseases as well as abnormal blood clotting. Additionally, several COVID-19 pneumonia patients had multiple kidney injuries, while other COVID-19 patients who passed away had severe renal damage [4,5]. SARS-CoV-2 infection results in acute kidney injury (AKI), which may lead to death [6,7]. Numerous renal pathologies, including acute tubular necrosis, proteinuria, hematuria, coagulopathy, and thrombosis problems, have been linked to COVID-19. The pathophysiological processes behind these clinical symptoms include a cytokine storm, hypoxemia, direct viral invasion via angiotensin-converting enzyme 2 and cathepsin L, electrolyte imbalance, and fever. The contributing factors for developing AKI have been evaluated in 161 intensive care unit (ICU) patients with a 28% incidence of AKI. AKI developed in approximately 35% of patients who had a history of chronic kidney disease (CKD). Furthermore, COVID-19 patients have organ failure and coagulopathy, resulting in a higher mortality rate [8]. COVID-19 also affects patients receiving chronic replacement therapies and those receiving kidney transplants [9,10].

According to the angiotensin converting enzyme 2 (ACE2) pathway, SARS-CoV-2 can directly infect kidney podocytes and proximal tubular cells and cause acute tubular necrosis, protein leakage in Bowman's capsule, collapsing glomerulopathy, and mitochondrial dysfunction. Other causes of AKI include cytokine storm, macrophage activation syndrome, and lymphopenia, which are all caused by SARS-CoV-2. Proposed mechanisms of kidney injury include sepsis-associated AKI and thrombotic disease [11]. The pathology of the kidneys involves collapsing segmental glomerulosclerosis, pigmented tubular casts, acute tubular damage, and glomerular fibrin thrombi [12].

Beyond 4 weeks after the onset of symptoms, the long-term complications of SARS-CoV-2 infection are further classified as sub-acute and chronic, or post-COVID-19 syndrome [13,14]. The possible renal effects of COVID-19 at 6 months after discharge have been emphasized by Huang and colleagues [15]. The COVID-19 vaccine has been administered in nearly all countries as one of the critical measures to control the pandemic. However, new-onset and relapsing glomerular diseases attributed to COVID-19 vaccination have emerged as a new source of concern [16]. Both the mRNA vaccine and the inactivated vaccine have the potential to cause new-onset and relapsing glomerular diseases; these diseases could occur after the first or second dose of vaccination [17,18]. The purpose of this review is to discuss the updated effects of COVID-19 on the kidney as well as investigate the manifestations caused by COVID-19, its long-term effects, and the impact of COVID-19 vaccination on kidney function.

## 2. COVID-19 and Manifestations of Kidney Dysfunction

Some of the clinical symptoms of COVID-19 include a cough, shortness of breath, muscle aches, disorientation, headache, sore throat, rhinorrhea, and chest pain [19,20]. AKI is more frequent in critically ill COVID-19 patients, according to recent data [21,22]. AKI is uncommon in individuals with mild-to-moderate COVID-19 (5%) [23]. The majority of the renal problems were subclinical in these patients [24]. The percentage of patients with AKI at any stage in a large cohort of 5449 hospitalized COVID-19 patients studied so far was 1993 (31.1%). A meta-analysis found a 1% pooled prevalence of CKD, with approximately 83.9% suffering from severe COVID-19 and a 53% mortality rate [25]. CKD is one of the most prevalent underlying illnesses in COVID-19 hospitalized patients [26,27].

### 2.1. Acute Tubular Necrosis

In China, 26 autopsies were examined using COVID-19 for the renal histopathology investigation. Acute tubular necrosis (ATN) was seen in every sample, and it was moderate to severe in 18 of the samples. Diffuse proximal tubule injury manifests as brush border loss, vacuolar degeneration, tubular lumen dilatation, epithelium detachment, and even frank necrosis. Only a few cells swelled in the distal tubules and collecting ducts. Using a transmission electron microscope, they discovered viral particles in podocytes and tubular epithelial cells (TECs) [28]. ATN was the most prevalent finding in the post-mortem examination of 42 individuals from the United States, with focal fibrin thrombi seen in 6 of the 42 autopsies, although the definitive virions could not be distinguished at the ultrastructural level [29]. Following a COVID-19 diagnosis, in three different case reports, Africans were shown to suffer collapsing focal segmental glomerulosclerosis and nephrotic syndrome (two men and one woman). In addition to the glomerular lesions, ATN lesions and mild-moderate interstitial infiltrates were seen in the kidney samples [30]. Based on endothelial cell damage, SARS-CoV-2-related kidney damage may potentially happen [31]. In histological analyses, endotheliosis of glomerular capillaries was also revealed in a postmortem examination, and viral particles were verified in the endothelial cells [32]. These conditions can cause microvascular dysfunction, which can lead to tissue inflammation, coagulopathy, and hypoxia. ATN is frequently discovered in autopsied kidney tissue; hence, SARS-CoV-2 may directly invade the hosts' renal tubular epithelial cells. Notably, AKI is frequently caused by tubular damage, which results in acute tubular necrosis [33,34].

*2.2. SARS-CoV-2 Infection and Acute Kidney Injury*

Acute kidney injury (AKI) is a clinical condition that develops when kidney function rapidly declines due to a number of causes. It is highly associated with morbidity and mortality [35]. Importantly, in previously reported SARS and MERS case reports, acute kidney injury (AKI) ranged from 3–43% of cases and had a high mortality rate (60–90%). Several studies have shown that people with COVID-19 have variable incidences of AKI [36]. AKI rates were significantly elevated to 14.5–50% in patients in the ICU who had a severe SARS-CoV-2 infection [37,38]. Serum creatinine levels were higher in AKI patients, as were leukocytosis, lymphopenia, thrombocytopenia, higher levels of D-dimer, and a prolonged activated partial thromboplastin time [39]. The patients also had increased procalcitonin levels and lactic dehydrogenase and aspartate aminotransferase activities [40]. The majority of AKI developed within 7 days, but it was much more severe and occurred much sooner in patients with higher baseline serum creatinine levels, whereas patients with normal baseline creatinine had a later onset of AKI and recovered quickly [41]. Patients with increased baseline serum creatinine also had significantly decreased estimated glomerular filtration rate (eGFR), elevated blood urea nitrogen (BUN), creatinine, high-grade proteinuria, and hematuria [42].

According to Li et al. [42], 28 patients were identified as having severe cases of COVID-19 disease, which was examined in 59 cases of inpatients from several hospitals in the Wuhan area. 63% of the patients had proteinuria, and many of them had protein found on the first day of admission, indicating the possibility of preexisting renal impairment. 27% of the patients had elevated BUN, and two of the patients who died also had higher BUN levels. 19% of the patients had elevated serum creatinine, and the patients who died had incredibly high amounts. Wang et al. [43] reported that when compared to patients who did not need ICU admission and to all hospital patients combined, the patients who needed ICU admission had higher BUN and creatinine levels at hospital admission. In COVID-19 patients, old age, male gender, high creatinine and blood urea nitrogen levels upon presentation, and a history of hypertension and diabetes were all risk factors for severe AKI [44]. Therefore, AKI may be a "typical" complication of a novel disease, and the rationale for treating AKI is the same as in other cases: it contributes to short-term morbidity and mortality, it can result in long-term complications like CKD, and it significantly raises both short- and long-term healthcare costs [45].

AKI is a broad term for a number of disorders that share a fast reduction in glomerular filtration rate as a characteristic [45]. AKI could also be caused by direct SARS-CoV-2 infection of renal tubular epithelial cells [28,46,47], and considering the participation of additional epithelia (lung, gastrointestinal tract, etc.), it appears to be a very plausible contributing factor in AKI [48]. AKI in COVID-19 patients, on the other hand, appears to be primarily caused by acute tubular injury, as in other forms of sepsis [49]. As with other causes of sepsis, some COVID-19 patients may develop macrophage activation syndrome with cytokine storms and elevated plasma ferritin levels [50,51]. AKI in COVID-19 is likely to have a variety of etiologies, such as severe hypoxemia, especially when combined with dehydration from high fevers, which, even in early-stage individuals, can be a challenge to the kidney. Fluid restriction and even diuretics, when mechanical ventilation is used in conjunction with positive end-expiratory pressure, may compromise renal perfusion while hypoxemia is reversed. Later, the condition worsens and hyperinflammation develops [45].

## 2.3. Proteinuria and Hematuria

Many studies looked at kidney function in patients with confirmed SARS-CoV-2 infections and discovered that proteinuria and hematuria were common [13,52]. Proteinuria is common during SARS-CoV-2 infection and has been reported in 7–63% of cases [53,54]. Cheng et al. (2020) reported hematuria in 26.7% of COVID-19 patients. Proteinuria is abundant or composed of albumin in some cases, indicating glomerular impairment. Proteinuria may have been caused by viral injury to podocytes as well as RAAS activation; angiotensin II accumulation may be responsible for nephron endocytosis and, as a result, increased glomerular permeability with proteinuria. This type of glomerulopathy has been associated with a number of illnesses, including viral infections [55]. Furthermore, increased hospital mortality is closely linked to both proteinuria and hematuria [56]. Since there were no prior values available when the patient was measured for proteinuria, the results should be interpreted with care. Nevertheless, these trials' subjects typically had conditions that increased their risk of kidney damage, including diabetes, hypertension, and obesity [40]. Additionally, proteinuria and hematuria should be interpreted with caution in critically ill, febrile, and oliguric patients, especially when determined by dipstick. Two black patients receiving treatment for SARS-CoV-2 infection were found to have two episodes of large proteinuria, severe AKI, and pathological collapsing glomerulopathy [57,58].

## 2.4. Thrombosis Complications

Several studies have noted the significant prevalence of acute thrombotic events in COVID-19 patients, particularly venous thrombosis and pulmonary embolism [59]. In comparison to survivors, non-survivors showed considerably greater levels of D-dimer and fibrinogen degradation products, as well as a longer prothrombin time upon admission. Additionally, during the last stages of hospitalization, significant reductions and lowerings in fibrinogen and antithrombin activity levels were seen in non-survivors, which is consistent with a clinical diagnosis of disseminated intravascular coagulation. In particular, angiotensin II is known to promote microvascular permeability, and the plasma levels of COVID-19 patients had an abnormally high level of angiotensin II that was directly correlated with viral load and lung damage [60,61], which stimulates platelets and causes tissue factor to be transcribed in endothelial cells. The fact that multiple complement system components can be released from endothelial cells in response to angiotensin II further supports the idea that endothelium plays a critical role in the pathogenesis of venous and arterial thrombosis in COVID-19 patients [62,63].

In COVID-19, especially in severe cases and in the late stages of the disease, the interaction between endothelium and immune cells could be extremely important. For example, the cytokine storm might cause an abrupt deterioration of the inflammatory response and hypercoagulation; as a result, the increased susceptibility of patients with cardiovascular diseases and/or diabetes may only reflect the effects of the underlying chronic inflammation and its response during SARS-CoV-2 infection [64]. In addition, there was a study of caduceus kidneys that revealed segmental microthrombi in the glomeruli of SARS-CoV-2 patients [65], and the kidney got damaged by the formation of extensive microthrombi. A new postmortem histopathologic investigation of 26 COVID-19 patients supports the potential of microthrombus formation because it revealed segmental fibrin thrombus formation in the glomerular capillary loops of three of the patients. Fibrin deposits in the glomerular loops of the kidney point to disruption of coagulation homeostasis, which can contribute to renal microcirculatory dysfunction and AKI [28]. Endothelial activation, which results in vasodilation, increased vascular permeability, and pro-thrombotic circumstances, can be brought on by microvascular inflammation. Additionally, it has been demonstrated that SARS-CoV-2 binds to platelets via ACE2, causing platelet activation and immunothrombosis [38]. As a result, platelet activation could have a role in the pathogenesis of COVID-19 AKI [66]. In conclusion, microvascular thrombosis is a significant cause of end-organ dysfunction, including respiratory failure and AKI, both of which are significant causes of mortality [67].

## 2.5. Effect of COVID-19 on Chronic Kidney Disease Patients

An abnormality in kidney structure or function that lasts longer than three months is referred to as chronic kidney disease (CKD) by the Kidney Disease Improving Global Outcomes (KDIGO) initiative. Based on glomerular filtration rate, KDIGO assigns a severity rating to the various stages of CKD [68]. CKD is a syndrome defined as persistent changes in kidney structure, function, or both, with consequences for the individual's health. In hospitalized patients, CKD is a well-known risk factor for AKI. Regardless of the underlying insult or disease, the most frequent pathological sign of CKD is renal fibrosis [69].

The majority of patients with CKD (in all stages) are over 65 years old; however, those who have the condition are more likely to advance to end-stage kidney disease (ESKD) if they are ≤65 years of age. The most prevalent underlying diseases linked to CKD are hypertension and diabetes mellitus [69]. According to the COVID-19 study, CKD was present in 0.5% to 37% of hospitalized patients and was linked to a significant co-morbidity burden and high 1-year mortality. Moreover, a study in Milan, Italy, investigated the early indicators of clinical outcomes [70]. Patients with COVID-19 who were hospitalized showed a significant prevalence of kidney involvement, which may also be related to previous chronic kidney disease. CKD per se is associated with a proinflammatory state, inferring that patients with CKD and COVID-19 might evolve with a more pronounced cytokinetic storm and hypercoagulability, which are important risk factors for AKI, severe illness, and mortality [40]. Depending on the series described, there were anywhere from 0.7 to 47.6% of patients with known CKD [48,71]. The mortality rate of the patients mentioned increased, and those with preexisting CKD had a greater prevalence of AKI [40,71]. Guidelines for the treatment of COVID-19 in patients with CKD are lacking. Although less than 10% of patients in this trial had an estimated glomerular filtration rate of 30 mL/min/1.73 m$^2$, dexamethasone was linked to both a decreased need for KRT and a lower mortality in all patients with severe COVID-19 [72]. Although several monoclonal antibody treatments have received emergency use authorization, their effectiveness in CKD patients has not been evaluated. Emergency use permission for the recombinant anti-IL-6 receptor antibody tocilizumab in patients with moderate-to-severe COVID-19 [73].

## 2.6. Effect of COVID-19 on ESKD Patients

Patients with ESKD who are intrinsically immune-compromised and have underlying comorbidities are particularly concerned about the COVID-19 epidemic [74]. Patients with ESKD typically present with fatigue and anorexia instead of the usual symptoms of cough and fever when compared to patients who are not on dialysis [75]. Patients with ESKD have a much higher risk of dying in the hospital than patients without ESKD, and this risk is exacerbated by older age and the requirement for mechanical ventilation [76]. Small cohorts with mortality rates between 14% and 30% provided the majority of the early reports on mortality. According to a more recent, larger cohort, patients with ESKD had a death rate of 31.7% compared to 25.4% for patients without the condition [77]. Since uremia is connected to reduced leucocyte function, patients with end-stage renal disease (ESRD) receiving hemodialysis (HD) or peritoneal dialysis (PD) may be more at risk [37]. The risk of COVID-19 is further increased by the fact that many patients also have co-morbid conditions such as diabetes, hypertension, and cardiovascular disease. The first information about HD patients using COVID-19 was released in China in March 2020 [43], and since then, other case series with varied death rates have been published.

## 2.7. COVID-19 and Kidney Transplant Recipients

Because of their vascular, anti-inflammatory, and immunomodulatory qualities, which give the kidney allograft immunological protection, corticosteroids were advocated for withdrawal or reduction of immunosuppressive therapy and for maintenance or introduction. While the optimal time to reintroduce immunosuppressive agents is unknown, graft rejection risk is increased by a sustained immunosuppressive reduction [78,79]. Furthermore, according to preliminary data, patients with COVID-19 kidney transplants exhibit

normal clinical signs, with fever and cough being frequent [80]. There are additional instances of patients having diarrhea and viral conjunctivitis in addition to a fever and cough. Given the fast decompensation seen in patients with ARDS, another crucial finding from early case reports relates to radiographic abnormalities with unilateral or bilateral infiltrates during admission of these patients, the majority of whom need ventilatory assistance [81].

### 3. Pathophysiology of COVID-19-Induced Kidney Dysfunction

The pathophysiology of COVID-19 AKI is considered to involve endothelial damage, activation of coagulation pathways, local and systemic inflammatory and immunological responses, and the renin-angiotensin system [82]. Cytokine storm, hypoxemia, direct viral invasion via angiotensin-converting enzyme 2 and cathepsin L, electrolyte imbalance, and fever are among the pathophysiological mechanisms underlying these clinical symptoms, which may also relate to renal injury and/or functional decline in the majority of seriously impacted patients [83,84].

### *3.1. Direct Viral Damage*

In situ hybridization and immunohistochemistry have been used to identify positive SARS-CoV-2 ribonucleic acid (RNA) polymerase gene fragments in kidney samples from SARS patients who passed away at autopsies. Additionally, MERS-CoV infections have been demonstrated to damage kidney epithelial cells by an apoptotic process controlled by receptors [85]. These investigations indicate that kidney epithelial cells are directly damaged by coronaviruses, causing cytotoxicity. Using autopsy immunohistochemistry, the kidney tissue of six patients was examined and found SARS-CoV-2 nucleocapsid (NP) protein in the kidney tubule, which may be associated with a possible direct tubular injury from the virus. A quantitative real-time polymerase chain reaction (qPCR) was used to find the SARS-CoV-2 RNA in the patients' urine. One effective receptor for SARS-CoV-2 infection has been discovered as ACE2 [86]. Although ACE2 is expressed in a variety of tissues, the kidney, cardiovascular tissues, and testis exhibit the highest levels of expression [87]. Thus, a direct viral infection may influence the mechanisms causing kidney damage.

#### 3.1.1. ACE2 Pathway

Transmembrane serine proteases (TMPRSS) cleave and activate the S protein once the SARS-CoV-2 spike (S) protein binds to ACE2 receptors, allowing the virus to release a fusion peptide that facilitates membrane fusion [88]. COVID-19-induced renal impairment may result from the synergistic interaction of immunological responses like cytokine storm, macrophage activation syndrome, and lymphopenia with virus-induced direct cytotropic effects. Multiorgan failure and probable volume depletion caused by decreased oral intake and a high temperature are two other potential explanations [89]. Drug toxicity, hemodynamic insult, endothelial dysfunction, hypercoagulability, sepsis, and lower oxygen delivery to the kidney may cause an ischemic injury to the kidney. It is vital to comprehend the basic molecular mechanisms and the pathophysiology of kidney damage and AKI in COVID-19 in order to create management plans and efficient therapeutic approaches [11].

#### 3.1.2. COVID-19 Kidney Dysfunction and Levels of ACE2 and TMPRSS2

It has been detected that ACE2 expression is high within proximal tubular cells but is not observed in immune cells of glomerular parietal epithelial cells. Furthermore, the data from single-cell RNA sequencing datasets of kidney tissues showed that the kidney's proximal tubular cells co-express ACE2 and transmembrane protease serine 2 (TMPRSS2) [90]. As a result, interaction between the virus and ACE2 on the cell surface is the first step in SARS-CoV-2 infection in humans. The receptor-binding domain of the viral spike protein is where ACE2 binds and interacts with external SARS-CoV-2. TMPRSS2 has been discovered as a protease responsible for the reaction, which is followed by the spike protein being cleaved by proteolytic means, enabling the union of cells [44]. However,

ACE2 downregulation is allegedly linked to a poor prognosis in severe acute respiratory syndrome coronavirus 1 (SARS-CoV-1) or SARS-CoV-2 infections [91]. The final result of ACE2 cleavage by metalloprotease 17, a disintegrin, and TMPRSS2 may act as a barrier to SARS-CoV-2 entrance. TMPRSS2, which promotes viral entrance by cleaving the S antigen into S1 (the active binding site), and ADAM17, which inhibits the production of ACE2 by shedding ACE2 proteins into a soluble form [92,93]. Reduced ACE2 expression in the human embryonic kidney cell line is the source of SARS-CoV-2 complications and end-organ damage and may hurt the host more severely through increased ACE2 toxic effects, such as the activation of proinflammatory cytokines [94].

### 3.1.3. ACE2 and the Renin-Angiotensin-Aldosterone System

Data from the Genotype-Tissue Expression Project show that the kidneys have significant levels of ACE2 expression [36]. The proximal tubule cells of the kidneys contain the highest concentration of ACE2, which is dispersed among many cells of the kidneys [95]. The presence of ACE2 in the cell membrane is necessary for SARS-CoV-2 to infect host cells [96]. ACE2 is a key modulator of the renin-angiotensin system (RAS) [97], a complex network of interconnected systems that controls both normal and abnormal heart, kidney, and pulmonary function. As the "entry door" for the SARS-CoV-2 virus, ACE2 receptors might be thought of as "devils" in the current pandemic. The epithelium of the renal tubules, however, appears to have been directly affected by viruses. The kidney has a significantly greater level of ACE2 RNA expression than the lung (almost 100-fold higher) [98]. This indicates that the kidney is vulnerable to viral infection and the harm that follows. In single-cell RNA sequencing data, podocytes and proximal tubular epithelial cells exhibit very high expression [88].

ACE2 functions as an enzyme within the renin-angiotensin system, metabolizing angiotensin II by cleaving a terminal peptide to generate angiotensin (1–7) (Ang 1–7) in addition to mediating SARS-CoV-2 entrance into cells [99]. Human ACE2 is thought to be downregulated after SARS-CoV-2 binds to it, which raises angiotensin II levels and lowers (Ang 1–7) [100]. It is currently unknown if an imbalance between angiotensin II and Ang 1–7 directly contributes to endothelial activation and COVID-19 AKI [101]. Inflammation and disruption of the glomerular filtration barrier are increased as a result of ACE2 depletion, which encourages the activation of AT1R by AT2R and amplifies the release of cytokines in response to oxidative stress. SARS-CoV-2 has been isolated from the urine of patients in Guandong [102] and Wuhan [43,102]. A larger case series also reported that the virus was present in one patient's urine, though it is unclear how many urine samples were examined [48]. However, Wang et al. were unable to detect the virus in 72 urine samples [43].

### 3.2. COVID-19 Cytokine Storm

To prevent viral replication and spread, the host must respond to viral infection by producing a variety of pro-inflammatory cytokines and activating CD4+ and CD8+ T lymphocytes [42]. Additionally, the cytopathic virus's tissue damage intensifies the inflammatory response by attracting nearby immune cells that produce a lot of pro-inflammatory cytokines, leading to further tissue injury [103]. The cytokine storm is significant in the immunopathology of COVID-19, just as it is in severe sepsis [42,104]. From the mild to the severe stages, COVID-19 disease was found to have varying levels of pro-inflammatory cytokines and chemokines [103]. In SARS-CoV-2-infected patients, a retrospective analysis revealed higher expression levels of IL-1$\beta$, IL-1RA, IL-7, IL-8, IL-10, IFN-G, monocyte chemoattractant peptide (MCP)-1, granulocyte-colony stimulating factor (G-CSF), macrophage inflammatory protein (MIP)-1A, MIP-1B, and tumor necrosis factor-alpha (TNF-$\alpha$) [105]. Furthermore, the severity of lung injury was strongly associated with increased plasma levels of IL-1$\alpha$, IL-1ra, IL-2, IL-7, IL-10, IL-17, IFN-G, inducible interferon protein (IP)-10, and G-CSF, which are positively related to SARS-CoV-2 viral titers. It is interesting to note that higher blood levels of IL-6 are linked to worse outcomes in AKI and

may serve as a useful biomarker for the early diagnosis and prediction of clinical outcomes like mortality and the requirement for dialysis. Numerous clinical investigations have shown that renal resident cells, such as podocytes, endothelial cells, mesangial cells, and tubular epithelial cells, can release IL-6 under specific circumstances. Meanwhile, immunological and inflammatory cells also actively react to IL-6 and contribute to the aggravation of renal damage in AKI and CKD [28]. In addition, lower levels of IFN-γ are associated with dysfunctional immune responses [106] and renal tubulointerstitial damage in CKD progression [107,108]. The cytokine storm damages renal cells by activating macrophages, causing erythrophagocytosis and anemia, capillary leak syndrome, and thrombosis, both of which are related to disseminated intravascular coagulation. Increased levels of the critical cytokine IL-6 cause shock, increased renal vascular permeability, and microcirculatory dysfunction by decreasing the expression of E-cadherin and activating the vascular endothelial growth factor [44].

### 3.3. Non-Specific Factors
Nephrotoxins

Additionally, the use of antibiotics, particularly in the midst of a critical illness, has been linked to a higher risk of AKI in COVID-19 patients. Examples of these antibiotics are vancomycin, colistin, and aminoglycosides [109]. Regarding the security of antivirals used to treat COVID-19, there are some unknowns. Remdesivir may cause mitochondrial damage in renal tubule epithelial cells, which would therefore have nephrotoxic effects. The likelihood of this kidney damage developing is highest at large dosages or after extended exposure [110]. The national clinical management protocol stated that remdesivir is contraindicated in patients with a GFR < 30 mL/min and in patients on hemodialysis [111]. An examination of the World Health Organization's international pharmacovigilance post-marketing databases identified a strong nephrotoxicity signal, showing a 20-fold increased risk of AKI while using remdesivir compared to other medications commonly used in COVD-19 (hydroxychloroquine, tocilizumab, and lopinavir/ritonavir) [112].

### 3.4. Hemodynamic Factors

In critically ill COVID-19 patients, AKI and cardiac damage are frequent and are linked to increased mortality. Kidney dysfunction in COVID-19 is also likely caused by cross-talk between the cardiovascular system and kidneys. Uncommon instances of acute myocarditis [113] and myocardial injury [114] have been described in patients with COVID-19, which might affect cardiac function, reduce cardiac output, congest renal veins, and compromise kidney perfusion [115]. Additionally, a rise in right heart pressure may make it easier for arterial and tissue hypoxia to deteriorate and the ventilation/perfusion mismatch to get worse [116]. The observed link between the risk of AKI and the use of vasopressors and mechanical ventilation further supports the idea that hemodynamic variables are a factor in COVID-19 AKI [117]. It is possible to develop more precise therapy and monitoring strategies by comprehending the interactions between the two organs in COVID-19 and the variables impacting organ functions.

### 3.5. Hypoxemia

The hypoxia-inducible factor system is extremely likely to be implicated because severe hypoxia is a defining feature of a severe SARS-CoV-2 infection. This could potentially affect the inflammatory response and outcome in both the kidneys and lungs [118]. Acute hypoxemia may affect kidney health and raise renal vascular resistance, which can lead to renal hypoperfusion and acute tubular damage [119,120]. Renal impairment may result from right-sided heart dysfunction and elevated venous pressure, which can increase interstitial and tubular hydrostatic pressure inside the encapsulated kidney and reduce net GFR and oxygen transport to the kidney [121]. AKI patients who required renal replacement therapy were almost universally on mechanical ventilation. According to these findings, kidney damage could still occur even in the absence of a direct virus attack on the organ if there is significant hypoxia, cytokine storms, or a combination of the two [118].

*3.6. Urinalysis and Electrolyte Imbalances*

Numerous investigations have discovered anomalies in urine sediment in addition to renal disease, which is indicated by an increase in serum creatinine. The severity of COVID-19 is linked to decreased serum sodium, potassium, and calcium concentrations, according to a pooled analysis [122,123]. Hypokalemia with enhanced kaliuresis, a sign of RAAS activation, was linked to the most severe SARS-CoV-2 infections that required ICU admission in a cohort of 175 COVID-19-infected individuals. In fact, 93% of ICU patients admitted to hospitals had hypokalemia at that time [38]. COVID-19 investigations have demonstrated that patients experience electrolyte problems, including imbalances in sodium, potassium, chloride, and calcium [124,125]. A case-control study revealed that electrolyte problems such as hyponatremia, hypokalemia, and hypochloremia were more prevalent in COVID-19 patients than in controls [126]. In fact, SARS-CoV-2-induced diarrhea, the use of diuretics, and other drug-induced tubulopathies may also be secondary causes of hypokalemia [8]. Electrolyte imbalances are caused by RAS alterations, gastrointestinal loss, the impacts of proinflammatory cytokines, and renal tubular failure caused by SARS-CoV-2 invasion [127,128].

RAS-inhibiting medications cause people with COVID-19 to produce less aldosterone, which can result in fluid and electrolyte imbalances. A disruption of the aldosterone system impairs colonic ion absorption and secretion, which leads to fluid and electrolyte imbalances [129]. Fluid and electrolyte abnormalities in some SARS-CoV-2-infected individuals have been linked to the syndrome of inappropriate antidiuretic hormone secretion (SIADH) [129].

COVID-19's subacute and long-term effects, which can damage many organ systems, are still being researched [130]. A similar cluster of persistent symptoms has been seen in survivors of earlier coronavirus infections, such as the SARS epidemic of 2003 and the MERS outbreak of 2012, raising concern for clinically significant COVID-19 aftereffects [111,131,132]. The development of an evidence-based multidisciplinary team approach for treating these individuals as well as the identification of research goals both require a comprehensive examination of sequelae following recovery from acute COVID-19. While the definition of the post-acute COVID-19 timeline is still being developed, it has been recommended that it should encompass the persistence of symptoms or the emergence of sequelae after 3 or 4 weeks have passed since the onset of acute COVID-19 symptoms [133,134], as replication-competent SARS-CoV-2 has not been isolated after 3 weeks [135]. It is further divided into two categories based on recent literature: (1) subacute or ongoing symptomatic COVID-19, which includes symptoms and abnormalities present from 4–12 weeks beyond acute COVID-19; and (2) chronic or post-COVID-19 syndrome, which includes symptoms and abnormalities persisting or present beyond 12 weeks of the onset of acute COVID-19 and not attributable to alternative diagnoses [133,136].

Early research on patients needing renal replacement therapy (RRT) revealed that between 27 and 64% were independent of dialysis by 28 days following ICU release [130,137]. At six months following the acute SARS-CoV-2 infection, 35% of patients reported a decreased estimated glomerular filtration rate (eGFR), and 13% of those patients experienced a new beginning of eGFR decline following the acute SARS-CoV-2 infection but with confirmed normal renal function [15]. The results of a group that was discharged after ICU admission and of which 10% required RRT are described by Hultström et al. [138].Ten percent of this cohort had CKD stage 3 or higher before admission. 16% (10/60) of the patients had advanced to a higher stage of CKD after three to six months, one of whom required dialysis [138]. With low rates of AKI (6%) and no mention of the need for RRT, Huang et al. (2021) present outcomes six months following diagnosis in a cohort of 1733 patients who were discharged from the hospital, of whom 4% were admitted to the ICU. 13% (107/822) of patients who had normal eGFR during the acute phase and no AKI at the time of their admission were later found to have a lower GFR. Planning and resource allocation for nephrology services depend heavily on an understanding of COVID-19's long-term effects on the kidney [15].

*3.7. COVID-19 Vaccination and Kidney Disease*

As of August 23, 2022, the WHO estimated that 12 billion doses of the COVID-19 vaccination had been given [139]. COVID-19 vaccines could prevent SARS-CoV-2 infection, symptomatic COVID-19, and severe COVID-19. In a real-world environment, COVID-19-related hospitalization, admission to the intensive care unit, and mortality were all successfully prevented with COVID-19 vaccines. Children have also shown the COVID-19 vaccine's protective effects [140]. However, it is unclear if COVID-19 vaccinations can stop long COVID-19 [139]. One or more COVID-like symptoms were more likely to be reported in the vaccinated group than in the unvaccinated group six weeks after the onset of illness, according to a cohort study of healthcare workers with confirmed COVID-19. Another study found that among healthcare professionals who did not need hospitalization, the number of vaccine doses was related to a decreased long-term COVID incidence [141]. However, one study found no significant difference between the vaccinated and unvaccinated groups in the mean number of post-acute sequelae of COVID-19 (PASC) symptoms reported each month throughout the follow-up period or in the probabilities of making a full recovery from PASC [142].

One of the key tactics to stop the COVID-19 pandemic has been rapid and widespread SARS-CoV-2 immunization. Live-replicating microbe-vectored vaccinations ought to be avoided since renal disease patients frequently have weakened immune systems. The mRNA vaccines BNT162b2 (Pfizer-BioNTech) and mRNA-1273 (Moderna), which are replication-defective viral-vectored vaccinations, are safe to use [143,144], indicating that the mRNA vaccines may more effectively produce protective immunity than ChAdOx1 nCoV-19. In keeping with the kidney, cases of acute kidney injury [145] and ANCA-associated vasculitis following vaccination [146] were reported after Pfizer-BioNTech vaccination. Patients who had acute renal injury and nephrotic syndrome with rapidly declining kidney function have been described in case reports after Sinovac-CoronaVac administration [147].

On the other hand, a number of studies point to the vaccine-induced anti-S IgG antibodies persisting for a longer time in hemodialysis patients, which would allay concerns about their rapid drop due to repeated dialysis over time. Patients maintained greater antibody levels despite the lengthy interval between getting the second dose of the vaccination and our research sample [148]. With respect to kidney safety, the Moderna vaccine has been associated with minimal de novo vasculitis [149] and ANCA glomerulonephritis. The pathogenesis of these vaccine-associated kidney diseases is possibly due to humoral and cellular immune responses [17].

3.7.1. Clinical Features of AKI and AKD after Vaccination

Over the past two years, hundreds of COVID-19 vaccine candidates have been created, examined, and finally released. AKI, AKD, proteinuria, edema, extensive hematuria, and other renal adverse effects necessitating hospitalization have, nevertheless, been reported in a small number of COVID-19 immunization case reports [150]. Most patients' serum creatinine levels (Scr) and proteinuria improved within three months of treatment. Most cases happened following mRNA vaccine and adenoviral vector injection, with a few reports of glomerulonephritis associated with inactivated virus vaccines [151]. There were 53 cases of AKI after SARS-CoV-2 vaccination as of February 2022. Only one case had a clear increase in Scr that resolved within 7 days of vaccination. Within 90 days, 37 patients' scores returned to baseline, and six patients had no results reported. Seven patients declined to answer [98].

There were a total of 53 cases, with 47 (89%) cases of new kidney involvement and 6 (11%) cases of relapse. The most common pathology (13 (25%): 11 new, 2 relapses) was minimal change disease (MCD), followed by IgA nephropathy (IgAN) (11 (21%): 9 new, 2 relapses), acquired thrombotic thrombocytopenic purpura (aTTP) (8 (15%): all new), and anti-neutrophil cytoplasmic autoantibodies (ANCA) vasculitis (8 (15%): all new). Four new cases of acute interstitial nephropathy (AIN), four new cases of membranous nephropathy (MN) (two new, two relapse), three new cases of anti-glomerular basement membrane (anti-

GBM) nephropathy, one new case each of granulomatous vasculitis and leukocytoclastic vasculitis, and four new cases of acute interstitial nephropathy (AIN). The participants were 64% male, and the average age was 58 years (standard deviation: 18) [98].

3.7.2. Insights into the Therapeutic Targeting Pathways of AKD following COVID-19 Vaccination

Renal impairment and/or failure may result from the direct and indirect effects of SARS-CoV-2 infection on the kidney, according to a number of theories [115]. Henry et al. [100] conducted a prospective study in 131 COVID-19 patients to investigate the role of complement as well as inflammatory and thrombotic parameters. They found that patients who experienced severe AKI and needed RRT while they were in the hospital had a large complement of activation and consumption. In each cohort, the C3a/C3 ratio was elevated in those who developed severe AKI and required RRT. Furthermore, in patients with severe AKI, there was a decrease in alternative and classical pathway activity, C4 consumption, and C3a elevation, indicating that in patients with COVID-19 thrombotic microangiopathy, this pathway represents a possible therapeutic target. CKD and other comorbidities, such as diabetes, have been linked to mortality in COVID-19 patients [75,152]. Patients with chronic dialysis for end-stage kidney disease (ESKD) or kidney transplant recipients (KTR) are also regularly identified as populations at high risk for a severe course of COVID-19 disease [153]. In order to improve clinical outcomes, these individuals must be managed clinically and therapeutically optimally. Despite the so-called "cytokine storm" being present in COVID-19 patients, Barbara et al. [154] reported a positive effect of tocilizumab (a humanized monoclonal antibody against the IL-6 receptor) administration in a kidney transplant recipient with proven SARS-CoV-2 pneumonia and ARDS who developed acute graft dysfunction in this research topic. Controlling immune system activation and the ensuing cytokine storm was the justification for its application in this situation. Days after receiving a single dose of tocilizumab, the patient experienced significant clinical (increased diuresis and oxygen saturation), laboratory (reduced white blood cell count, C-reactive protein, and IL-6), and radiological (reduced pulmonary opacities) improvements. This shows the significance of immunomodulation in COVID-19 disease, particularly in this particular subgroup of patients. Moreover, AKI may worsen the inflammatory state of these critically ill patients since so many inflammatory mediators are both removed and created by kidney-resident cells that are immunologically active.

## 4. Limitations

COVID-19 has been known to affect various organs, including the kidneys. While studies have provided valuable insights into the potential implications of COVID-19 for kidney dysfunction, it is important to acknowledge some limitations. Firstly, many studies are based on observational data, which can be influenced by confounding factors and biases. It can be challenging to establish a direct causal relationship between COVID-19 and kidney dysfunction due to the presence of other comorbidities and variations in patient populations. Additionally, the rapidly evolving nature of the pandemic has led to a significant amount of research being conducted in a short span of time. While this is commendable, it also means that some studies may have limited sample sizes or insufficient follow-up periods, making it difficult to draw definitive conclusions. Furthermore, the heterogeneity in diagnostic criteria, treatment protocols, and severity classification across studies can impact the consistency and comparability of findings. Variations in study designs, population demographics, and healthcare resources also contribute to the limitations of generalizing the results. To overcome these limitations, larger and more rigorous studies, including randomized controlled trials, would be beneficial. Collaborative efforts, data sharing, and standardized protocols could help enhance the quality and reliability of research on COVID-19 and its potential implications for kidney dysfunction.

## 5. Conclusions

There is an association between COVID-19 and the kidney due to the high expression of ACE2 in kidney tissue. SARS-CoV-2 infection can increase mortality as well as cause new kidney damage, making it more difficult to treat and care for those who already have kidney disorders. In COVID-19 instances, renal dysfunction, primarily AKI, haematuria, and proteinuria, occurs; however, kidney problems are linked to increased mortality. The pathophysiology of AKI can be associated with COVID-specific mechanisms such as direct viral entry, proinflammatory cytokines provoked by the viral infection, electrolyte imbalance, and hypoxia, as mentioned in Figure 1. Direct viral-induced tubular or glomerular damage, sepsis-related renal dysfunction, and thrombotic disease are a few proposed pathways for kidney damage. Acute tubular injury, glomerular fibrin thrombi, pigmented tubular casts, and collapsing localized segmental glomerulosclerosis are all examples of kidney pathology. Similarly, it is clear that some of the COVID-19 patients' approved drugs had an impact on renal function and needed to be delivered with great caution. However, some of the current vaccinations are associated with a slight increase in kidney injury. Given the crucial role kidneys play in controlling blood pressure and cleansing the blood of harmful substances, kidney safety in COVID-19 patients continues to be of the utmost concern. Notably, in order to develop novel medications to treat kidney impairment in SARS-CoV-2-infected patients, it is imperative to understand the biochemical pathways and essential targeted molecules.

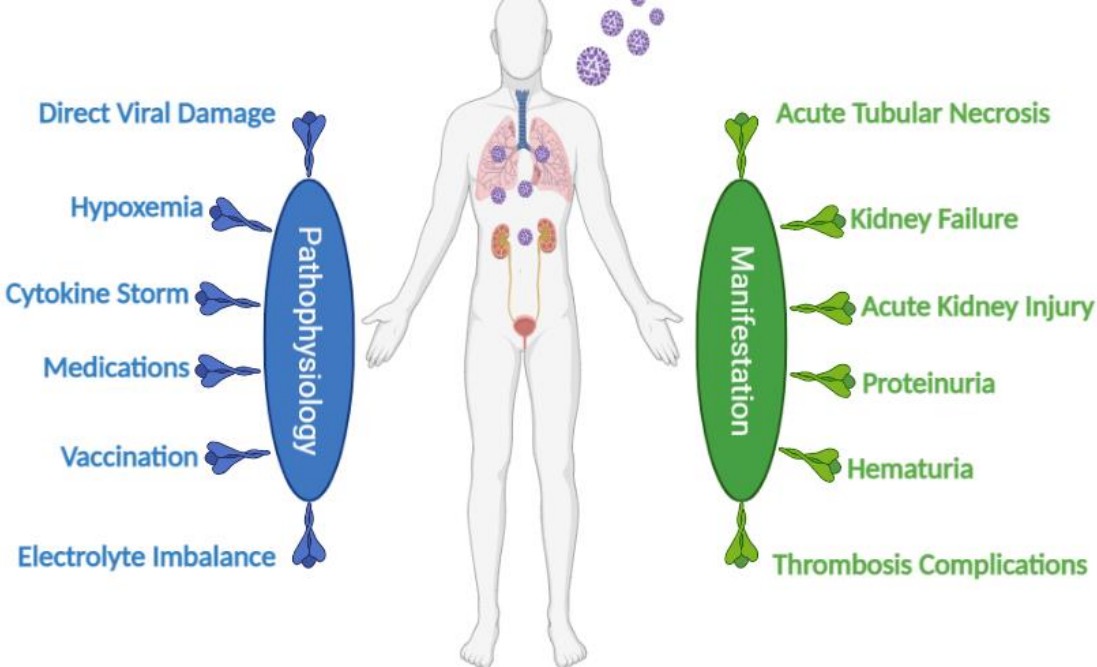

**Figure 1.** An illustration presenting a summarization of the most probable manifestations and pathophysiological mechanisms of COVID-19-induced kidney dysfunction.

**Author Contributions:** Conceptualization, A.A.-M. and E.H.B.; writing—original draft preparation, A.A.-M., E.H.B. and M.Y.Z.; writing—review and editing, M.Y.Z. All authors have read and agreed to the published version of the manuscript.

**Funding:** This research received no external funding.

**Institutional Review Board Statement:** Not applicable.

**Informed Consent Statement:** Not applicable.

**Data Availability Statement:** Not applicable.

**Conflicts of Interest:** The authors declare no conflict of interest.

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
