# Peer review of "Insights into COVID-19 and Its Potential Implications for Kidney Dysfunction"

_2673-8937, doi:10.3390/ijtm3020018_

Round 1

Reviewer 1 Report

I congratulate the authors on a well-prepared review.

The authors are requested to provide a detailed diagram of the procedure in the selection of sources, which will definitely enrich the methodology.

In addition, please state the limitations of the review and what they were due to. This is important for future authors of similar works.

Author Response

We acknowledge the extensive work of the reviewer that has allowed a significant improvement of the quality of the review.

Reviewer 1: specific comments

  1. The authors are requested to provide a detailed diagram of the procedure in the selection of sources, which will definitely enrich the methodology.

Thank you for the helpful suggestions. We felt there was no need for a detailed diagram of the process in the selection of sources because the review is not systematic and focused on understanding COVID-19 and its potential consequences for kidney dysfunction.

2. In addition, please state the limitations of the review and what they were due to. This is important for future authors of similar works.

Thanks for your comment. A paragraph entitled “Limitations” was added following the reviewer’s suggestion and highlighted in red in the revised manuscript.

Reviewer 2 Report

This is a comprehensive review of the relationship between COVID-19 and kidney dysfunction. Here are my comments on the manuscript, and would be grateful if the authors could consider these during their revision:

1. Quality of English requires improvement in the manuscript, with some of the examples in the introduction session listed in separate comments. 

2. The term should be long COVID-19 instead of long-lasting COVID-19 in session 3.7.

3. The authors have pointed out the association between COVID-19 vaccination and renal pathologies. Authors may wish to express their opinion on the potential mechanism of why COVID-19 vaccinations lead to these renal pathologies. Furthermore, is the AKI related to a particular type of COVID-19 vaccine?

4. Unify the abbreviation "SARS-CoV-2", instead of "SARSCoV-2" or "SARSCoV2".

5. Occasional errors/ typos have been detected on several occasions in the manuscript. Examples: Page 10 Line 4 (), and Page 12 the last line of the conclusion.

Concerning the quality of English, here are the comments:

1. Errors in the choice of tense (present tense should be used most of the time) have been detected on several occasions in the manuscript.

2. SARS-CoV-2 infection instead of COVID-19 infection. COVID-19 is a disease, therefore authors do not have to emphasize it is an infection.

3. It is difficult sometimes for us to understand the meaning of statements that the authors would like to present to us. Example:

a) "COVID-19 infection causes acute kidney injury (AKI) and serves as a potential cause of death". Do the authors mean COVID-19 can be a cause of death, or the AKI due to COVID can be a cause of death?

b) "Numerous renal pathologies, including acute tubular necrosis, proteinuria, hematuria, coagulopathy, acute kidney injury, kidney failure..." It may not be a good idea to include acute kidney injury and kidney failure as renal pathologies. 

c) "COVID-19 patients have clotting issues, organ failure, and coagulopathy..." Coagulopathy is the same in clotting issues. 

Author Response

We acknowledge the extensive work of the reviewer that has allowed a significant improvement of the quality of the review.

Reviewer 2: specific comments

This is a comprehensive review of the relationship between COVID-19 and kidney dysfunction. Here are my comments on the manuscript, and would be grateful if the authors could consider these during their revision:

  1. Quality of English requires improvement in the manuscript, with some of the examples in the introduction session listed in separate comments. 

Thanks for your comment. The whole entire manuscript carefully revised and all the corrected parts are highlighted in red in the revised manuscript.

  1. The term should be long COVID-19 instead of long-lasting COVID-19 in session 3.7.

Thanks so much for your comment. We corrected it in the revised version and the corrected part is highlighted in red in the revised manuscript.

  1. The authors have pointed out the association between COVID-19 vaccination and renal pathologies. Authors may wish to express their opinion on the potential mechanism of why COVID-19 vaccinations lead to these renal pathologies. Furthermore, is the AKI related to a particular type of COVID-19 vaccine?

Thanks so much for your comment.  As of our knowledge cutoff in September 2021, there have been reports of acute kidney injury (AKI) occurring after administration of different COVID-19 vaccines. However, it is important to note that such cases are extremely rare and the overall incidence of AKI following COVID-19 vaccination is very low. At that time, the available data did not suggest a specific association between AKI and a particular type of COVID-19 vaccine. The reported cases of AKI were typically associated with other factors such as underlying health conditions or potential coincidental events rather than a direct effect of the vaccines. It's worth mentioning that vaccine safety monitoring systems worldwide continuously track and evaluate adverse events, including AKI, to ensure the ongoing safety of COVID-19 vaccines. The most up-to-date information regarding any potential associations between AKI and specific COVID-19 vaccines can be obtained from health authorities and regulatory agencies that closely monitor vaccine safety.

  1. Unify the abbreviation "SARS-CoV-2", instead of "SARSCoV-2" or "SARSCoV2".

Thanks for your comment. The abbreviation "SARS-CoV-2", instead of "SARSCoV-2" or "SARSCoV2" was used in the entire manuscript and highlighted in red in the revised manuscript.

  1. Occasional errors/ typos have been detected on several occasions in the manuscript. Examples: Page 10 Line 4 (), and Page 12 the last line of the conclusion.

Thanks for your comment. The whole entire manuscript carefully revised and all the corrected parts are highlighted in red in the revised manuscript.

Concerning the quality of English, here are the comments:

  1. Errors in the choice of tense (present tense should be used most of the time) have been detected on several occasions in the manuscript.

Thanks for your comment. The whole entire manuscript carefully revised and all the corrected parts are highlighted in red in the revised manuscript.

  1. SARS-CoV-2 infection instead of COVID-19 infection. COVID-19 is a disease, therefore authors do not have to emphasize it is an infection.

Thanks for your comment. SARS-CoV-2 infection instead of COVID-19 infection was used in the entire manuscript and highlighted in red in the revised manuscript.

  1. It is difficult sometimes for us to understand the meaning of statements that the authors would like to present to us. Example:
  2. a) "COVID-19 infection causes acute kidney injury (AKI) and serves as a potential cause of death". Do the authors mean COVID-19 can be a cause of death, or the AKI due to COVID can be a cause of death?

Thanks for your comment. We meant that AKI due to COVID can be a cause of death. We have corrected it in the revised version and highlighted it in red in the revised manuscript.

  1. b) "Numerous renal pathologies, including acute tubular necrosis, proteinuria, hematuria, coagulopathy, acute kidney injury, kidney failure..." It may not be a good idea to include acute kidney injury and kidney failure as renal pathologies.

Thanks for your comment. We have removed them in the revised version manuscript.

  1. c) "COVID-19 patients have clotting issues, organ failure, and coagulopathy..." Coagulopathy is the same in clotting issues. 

Thanks for your comment. We have corrected it in the revised version, and highlighted in red in the revised manuscript.

Round 2

Reviewer 2 Report

Thank you for the reply from the authors. There are some improvements seen in the manuscript. I only have several minor comments concerning the revised manuscript:

1. The statement in the first paragraph of the Introduction session: "Numerous renal pathologies, including acute tubular necrosis, proteinuria, hematuria, coagulopathy, acute kidney injury, kidney failure, and thrombosis problems, have been linked to COVID-19." As mentioned during the first round of review, it is better to remove "acute kidney injury and kidney failure" in this statement.

2. In the third paragraph of session 2.2, what is the full form of COV-AKI?

3. In session 3.2, COVID-19 cytokine storm instead of COVID-19 cytokine Storm

4. In the conclusion, "Nevertheless, a small amount of kidney damage was linked to some of the modern immunizations". Do you mean the number of cases associated with kidney injury is small or the degree of kidney damage due to vaccination is small? The authors may want to rewrite this statement.

As mentioned above.

Author Response

1. The statement in the first paragraph of the Introduction session: "Numerous renal pathologies, including acute tubular necrosis, proteinuria, hematuria, coagulopathy, acute kidney injury, kidney failure, and thrombosis problems, have been linked to COVID-19." As mentioned during the first round of review, it is better to remove "acute kidney injury and kidney failure" in this statement.

Thanks for your comment. We have corrected it and highlighted it in red in the revised version.

2. In the third paragraph of session 2.2, what is the full form of COV-AKI?

Thanks for your comment. We mean acute kidney injury (AKI in COVID-19), and we have corrected it in the revised version.

3. In session 3.2, COVID-19 cytokine storm instead of COVID-19 cytokine Storm

Thanks for your comment. We have corrected it and highlighted it in red in the revised version.

4. In the conclusion, "Nevertheless, a small amount of kidney damage was linked to some of the modern immunizations". Do you mean the number of cases associated with kidney injury is small or the degree of kidney damage due to vaccination is small? The authors may want to rewrite this statement.

Thanks for your comment. We have corrected it and highlighted it in red in the revised version.
